# Manifold Density Estimation via Generalized Dequantization

**James A. Brofos** [1]   **Marcus A. Brubaker** [2,3]   **Roy R. Lederman** [1]

## Abstract

Density estimation is an important technique for characterizing distributions given observations. Much existing research on density estimation has focused on cases wherein the data lies in a Euclidean space. However, some kinds of data are not well-modeled by supposing that their underlying geometry is Euclidean. Instead, it can be useful to model such data as lying on a *manifold* with some known structure. For instance, some kinds of data may be known to lie on the surface of a sphere. We study the problem of estimating densities on manifolds. We propose a method, inspired by the literature on "dequantization," which we interpret through the lens of a coordinate transformation of an ambient Euclidean space and a smooth manifold of interest. Using methods from normalizing flows, we apply this method to the dequantization of smooth manifold structures in order to model densities on the sphere, tori, and the orthogonal group.

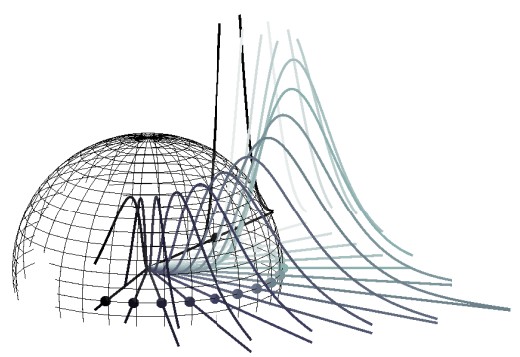

*Figure 1.* We model densities on a manifold as a projection, or "quantization," onto the manifold from an ambient Euclidean space. To enable density computations we use a "dequantization density" which can depend position on the manifold. In this figure the manifold in question is $\mathbb{S}^2$, embedded in $\mathbb{R}^3$ and the dequantization density, illustrated here for a set of locations along the equator of $\mathbb{S}^2$, is over $r \in \mathbb{R}_+$, the distance from the origin in the direction $y \in \mathbb{S}^2$. Density on the manifold can be estimated via importance sampling by marginalizing over $\mathbb{R}^+$ for a given $y \in \mathbb{S}^2$ using the dequantization distribution as an importance distribution.

## 1. Introduction

This material appears in greater detail in our long-form version on arXiv; please see Brofos et al. (2021) for full details.

Certain kinds of data are not well-modeled under the assumption of an underlying Euclidean geometry. Examples include data with a fundamental directional structure, data that represents transformations of Euclidean space (such as rotations and reflections), data that has periodicity constraints or data that represents hierarchical structures. In such cases, it is important to explicitly model the data as lying on a *manifold* with a suitable structure; for instance

[1]Department of Statistics and Data Science, Yale University [2]Department of Electrical Engineering and Computer Science, York University, Toronto, Canada [3]Vector Institute, Toronto, Canada. Correspondence to: James A. Brofos <james.brofos@yale.edu>.

Third workshop on *Invertible Neural Networks, Normalizing Flows, and Explicit Likelihood Models* (ICML 2021). Copyright 2021 by the author(s).

a sphere would be appropriate for directional data, the orthogonal group for rotations and reflections, and the torus captures structural properties of periodicity.

The contribution of this work is to express density estimation on manifolds as a form of dequantization. Given a probability density in an ambient Euclidean space, one can obtain the density on the manifold by performing a *manifold change-of-variables* in which the manifold structure appears and then projecting out any auxiliary structures. This marginalization can be viewed as analogous to "quantization" where, for instance, continuous values are discarded and only rounded integer values remain. In this view the auxiliary structure defines how the manifold could be "dequantized" into the ambient Euclidean space. By marginalizing along these auxiliary dimensions, one obtains the marginal distribution on the manifold. In practice, however, one has only the manifold-constrained observations from an unknown distribution on the manifold. A second contribution of this work is to formulate the density estimation as a learning problem on the ambient Euclidean space. We show how to invoke the manifold change-of-variables, and then perform the marginalization along the auxiliary dimensions to ob-

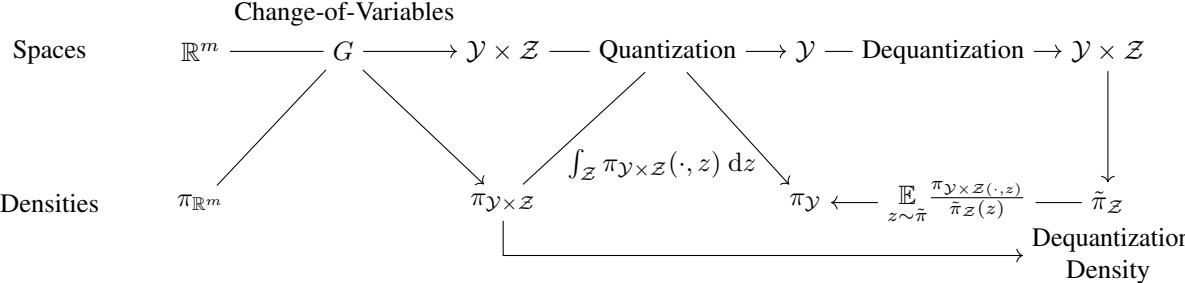

*Figure 2.* The dequantization roadmap. In the first row, we begin with $\mathbb{R}^m$ (or a space identical to $\mathbb{R}^m$ up to a set of measure zero). This Euclidean space can be transformed into the product of manifolds $\mathcal{Y} \times \mathcal{Z}$ via a change-of-variables $G : \mathbb{R}^m \to \mathcal{Y} \times \mathcal{Z}$. Quantization takes the product manifold $\mathcal{Y} \times \mathcal{Z}$ to its $\mathcal{Y}$-component alone. In the second row, we may begin with a probability density $\pi_{\mathbb{R}^m}$ defined on $\mathbb{R}^m$. Under the change-of-variables $G$ we obtain a new probability density $\pi_{\mathcal{Y} \times \mathcal{Z}}$ which is related to $\pi_{\mathbb{R}^m}$ by the manifold change-of-variables eq. (1). Quantizing $\mathcal{Y} \times \mathcal{Z}$ marginalizes out the $\mathcal{Z}$-component of $\pi_{\mathcal{Y} \times \mathcal{Z}}$. We may equivalently introduce a dequantization density $\tilde{\pi}_{\mathcal{Z}}$ and compute the marginal density on $\mathcal{Y}$ via importance sampling.

*Table 1.* Table of the matrix manifold dequantizations considered in this work. We show the corresponding auxiliary structure, the dequantization transformation, the resulting Euclidean space, and the Jacobian determinant of the transformation.

| Manifold | Auxiliary Structure | Euclidean Space | Dequantization | Jacobian Determinant |
|---|---|---|---|---|
| $\mathbb{S}^{m-1}$ | $\mathbb{R}_+$ | $\mathbb{R}^m$ | Spherical coordinates $(y, r) \mapsto ry$ | $r^{m-1}$ |
| $\mathbb{T}^m$ | $\mathbb{R}_+ \times \cdots \times \mathbb{R}_+$ | $\mathbb{R}^{2m}$ | Iterated polar coordinates $(y_i, r_i) \mapsto r_i y_i$ | $\prod_{i=1}^m r_i$ |
| $\text{Stiefel}(m, n)$ | $\text{Tri}_+(n)$ | $\mathbb{R}^{m \times n}$ | QR decomposition $(\mathbf{Y}, \mathbf{R}) \mapsto \mathbf{YR}$ | $\mathbf{R}_{11}^{m-1} \cdots \mathbf{R}_{nn}^{m-n}$ |
| $\text{Stiefel}(m, n)$ | $\text{PD}(n)$ | $\mathbb{R}^{m \times n}$ | Matrix polar decomposition $(\mathbf{Y}, \mathbf{R}) \mapsto \mathbf{YR}$ | Automatic differentiation |

tain effective estimates of the density on the manifold. An advantage of our dequantization approach is that it *allows one to utilize any expressive density directly on the ambient Euclidean space* (e.g., RealNVP (Dinh et al., 2017), neural ODEs (Chen et al., 2018; Grathwohl et al., 2018) or any other normalizing flow (Kobyzev et al., 2020)); the dequantization approach does not require a practitioner to construct densities intrinsically on the manifold.

## 2. Theory

**Theorem 1.** Let $\mathcal{Y}$ and $\mathcal{Z}$ be smooth manifolds embedded in $\mathbb{R}^n$ and $\mathbb{R}^p$, respectively. Let $G : \mathbb{R}^m \to \mathcal{Y} \times \mathcal{Z}$ be a smooth, invertible transformation. Let $\pi_{\mathbb{R}^m}$ be a density on $\mathbb{R}^m$. Under the change-of-variables $G$, the corresponding density on $\mathcal{Y} \times \mathcal{Z}$ is given by,

$$\pi_{\mathcal{Y} \times \mathcal{Z}}(y, z) = \frac{\pi_{\mathbb{R}^m}(x)}{\sqrt{\det(\nabla G(x)^\top \nabla G(x))}} \quad (1)$$

where $x = G^{-1}(y, z)$.

Even when $G$ is not an invertible mapping, it may be possible to compute the change-of-variables when $G$ is invertible

on partitions of $\mathbb{R}^m$.

**Corollary 1.** Let $\mathcal{O}_1, \ldots, \mathcal{O}_l$ be a partition of $\mathbb{R}^m$. Let $G : \mathbb{R}^m \to \mathcal{Y} \times \mathcal{Z}$ be a function and suppose that there exist smooth and invertible functions $G_i : \mathcal{O}_i \to \mathcal{Y} \times \mathcal{Z}$ such that $G_i = G|\mathcal{O}_i$ for $i = 1, \ldots, l$. Then, if $x \sim \pi_{\mathbb{R}^m}$, the density of $(y, z) = G(x)$ is given by

$$\pi_{\mathcal{Y} \times \mathcal{Z}}(y, z) = \sum_{i=1}^l \frac{\pi_{\mathbb{R}^m}(x_i)}{\sqrt{\det(\nabla G_i(x_i)^\top \nabla G_i(x_i))}}. \quad (2)$$

where $x_i = G_i^{-1}(y, z)$.

How does theorem 1 relate to the dequantization of smooth manifolds?

### 2.1. Dequantization

Manifolds of interest (such the sphere, the torus, or the orthogonal group) can be introduced as elements of a new coordinate system for an ambient Euclidean space. In each case, the manifold appears with an auxiliary manifold which may not be of immediate interest. Namely, (i) The sphere appears with set of positive real numbers when defining

a coordinate system for $\mathbb{R}^m \setminus \{0\} \cong \mathbb{S}^{m-1} \times \mathbb{R}_+$; (ii) The torus appears the product manifold of $m$ copies of the positive real numbers when defining a coordinate system for $\mathbb{R}^{2m} \setminus \{0\} \cong \mathbb{T}^m \times \mathbb{R}_+ \times \ldots \times \mathbb{R}_+$; (iii) the Stiefel manifold appears with the set of lower-triangular matrices with positive diagonal entries when defining a coordinate system of full-rank matrices: $\mathrm{FR}(n, p) \cong \mathrm{Stiefel}(n, p) \times \mathrm{Tri}_+(p)$. We would like to marginalize out these "nuisance manifolds" so as to obtain distributions on the manifold of primary interest. A convenient means to achieve this is to introduce an importance sampling distribution over the nuisance manifold. Formally, we have the following result, which is an immediate consequence of theorem 1.

**Corollary 2.** Let $\mathcal{Y}$, $\mathcal{Z}$, $G$, and $\pi_{\mathcal{Y} \times \mathcal{Z}}$ be as defined in theorem 1. Let $\tilde{\pi}_{\mathcal{Z}}$ be a non-vanishing density on $\mathcal{Z}$. To obtain the marginal density on $\mathcal{Y}$, it suffices to compute,

$$\pi_{\mathcal{Y}}(y) = \underset{z \sim \tilde{\pi}_{\mathcal{Z}}}{\mathbb{E}} \frac{\pi_{\mathcal{X}}(x)}{\tilde{\pi}_{\mathcal{Z}}(z) \cdot \sqrt{\det(\nabla G(x)^\top \nabla G(x))}}, \quad (3)$$

where $x = G^{-1}(y, z)$.

## 3. Discussion

We investigate the problem of density estimation given observations on a manifold using the dequantization procedure described in section 2.

**Problem.** Let $\mathcal{Y}$ be a manifold embedded in $\mathbb{R}^n$ and let $\pi_{\mathcal{Y}}$ be a density on $\mathcal{Y}$. Given observations of $\pi_{\mathcal{Y}}$, construct an estimate $\hat{\pi}_{\mathcal{Y}}$ of the density $\pi_{\mathcal{Y}}$. We apply eq. (3) in order to obtain the density estimate on $\mathcal{Y}$. Generating samples from $\pi_{\mathcal{Y}}$ may be achieved by first sampling $x \sim \pi_{\mathcal{X}}$, applying the transformation $G(x) = (y, z)$, and taking $y$ as a sample from the approximated distribution $\hat{\pi}_{\mathcal{Y}}$.

### 3.1. Densities on $\mathbb{R}^m$

As $\mathbb{R}^m$ is a Euclidean space, we have available a wealth of possible mechanisms to produce flexible densities in the ambient space. One popular choice is RealNVP (Dinh et al., 2017). An alternative is neural ODEs which parameterizes a vector field in the Euclidean space; the change in probability density under the vector field flow is obtained by integrating the instantaneous change-of-variables formula (Chen et al., 2018; Grathwohl et al., 2018).

### 3.2. Objective Functions

We consider two possible objective functions for density estimation. The first is the evidence lower bound of the observations $\{y_1, \ldots, y_{n_{\mathrm{obs}}}\}$:

$$\log \hat{\pi}_{\mathcal{Y}}(y_i) \geq \underset{z \sim \tilde{\pi}_{\mathcal{Z}}}{\mathbb{E}} \log \frac{\pi_{\mathbb{R}^m}(G^{-1}(y_i, z))}{\tilde{\pi}_{\mathcal{Z}}(z) \cdot \sqrt{\det(\nabla G(x)^\top \nabla G(x))}}. \quad (4)$$

This follows as a consequence of Jensen's inequality applied to eq. (3). Experimental results using this objective function are denoted with the suffix (ELBO). The second is the log-likelihood computed via importance sampling:

$$\log \hat{\pi}_{\mathcal{Y}}(y_i) = \log \underset{z \sim \tilde{\pi}_{\mathcal{Z}}}{\mathbb{E}} \frac{\pi_{\mathbb{R}^m}(G^{-1}(y_i, z))}{\tilde{\pi}_{\mathcal{Z}}(z) \cdot \sqrt{\det(\nabla G(x)^\top \nabla G(x))}}. \quad (5)$$

Because the calculation of eq. (5) requires an importance sampling estimate, experimental results using this objective function are denoted with the suffix (I.S.).

## 4. Experimental Results

To demonstrate the effectiveness of the approach, we now show experimental results for density estimation on three different manifolds: the sphere, the torus and the orthogonal group. In our comparison against competing algorithms, we ensure that each method has a comparable number of learnable parameters. Our evaluation metrics are designed to test the fidelity of the density estimate to the target distribution. In all of our examples we use rejection sampling in order to draw samples from the target distribution.

### 4.1. Sphere and Hypersphere

Our first experimental results concern the sphere $\mathbb{S}^2$ where we consider a multimodal distribution with four modes. We consider performing density estimation using the ELBO (eq. (4)) and log-likelihood objective functions (eq. (5)); we construct densities in the ambient space using RealNVP and neural ODEs. As baselines we consider the Möbius transform approach described in (Rezende et al., 2020), which is a specialized normalizing flow method for tori and spheres, and the neural manifold ODE applied to the sphere as described in (Lou et al., 2020). We give a comparison of performance metrics between these methods in table 2. In these experiments, we find that parameterizing a neural ODE model in the ambient space gave the better KL-divergence and effective sample size (ESS) metrics than RealNVP when our dequantization approach is used. We find that our dequantization algorithm minimizing either eq. (4) or eq. (5) achieves similar performance in the first and second moment metrics. However, when using eq. (5), slightly lower KL-divergence metrics are achievable as well as slightly larger effective sample sizes. In either case, dequantization tends to outperform the Möbius transform on this multimodal density on $\mathbb{S}^2$. The manifold ODE method is outperformed by the ODE dequantization algorithms with both eq. (4) and eq. (5).

We next consider a multimodal density $\mathbb{S}^3 \cong \mathrm{SU}(3)$ (the special unitary group). As before, we compare dequantization to Möbius flow transformations and manifold neural ODEs and present results in table 3. Similar to the case of the

*Table 2.* Comparison of dequantization to normalizing flows on the multimodal density on $\mathbb{S}^2$. Averages were computed using ten random trials for the dequantization procedures and eight random trials for the normalizing flow (because two random trials exhibited divergent behavior and were excluded). The dequantization procedure is illustrated for both the ELBO loss and the KL divergence loss.

| Method | Mean MSE | Covariance MSE | KL$(q\|p)$ | KL$(p\|q)$ | Relative ESS |
|---|---|---|---|---|---|
| Deq. ODE (ELBO) | $0.0012 \pm 0.0002$ | $0.0006 \pm 0.0001$ | $0.0046 \pm 0.0002$ | $0.0046 \pm 0.0002$ | $99.0990 \pm 0.0401$ |
| Deq. ODE (I.S.) | $0.0014 \pm 0.0002$ | $0.0010 \pm 0.0001$ | $0.0029 \pm 0.0001$ | $0.0029 \pm 0.0001$ | $99.4170 \pm 0.0225$ |
| Deq RealNVP (ELBO) | $0.0004 \pm 0.0001$ | $0.0003 \pm 0.0001$ | $0.0231 \pm 0.0010$ | $0.0212 \pm 0.0009$ | $95.9540 \pm 0.1688$ |
| Deq. RealNVP (I.S.) | $0.0005 \pm 0.0002$ | $0.0002 \pm 0.0000$ | $0.0124 \pm 0.0006$ | $0.0115 \pm 0.0006$ | $97.8240 \pm 0.1183$ |
| Man. ODE | $0.0010 \pm 0.0004$ | $0.0009 \pm 0.0002$ | $0.0085 \pm 0.0007$ | $0.0083 \pm 0.0007$ | $98.3860 \pm 0.1328$ |
| Möbius | $0.0021 \pm 0.0005$ | $0.0019 \pm 0.0005$ | $0.0595 \pm 0.0025$ | — | $89.2575 \pm 0.4888$ |

*Table 3.* Comparison of dequantization to normalizing flows on the multimodal density on $\mathbb{S}^3$. Averages were computed using ten random trials for the dequantization procedures and nine random trials for the normalizing flow (one random trial exhibited divergent behavior and was excluded).

| Method | Mean MSE | Covariance MSE | KL$(q\|p)$ | KL$(p\|q)$ | Relative ESS |
|---|---|---|---|---|---|
| Deq. ODE (ELBO) | $0.0009 \pm 0.0001$ | $0.0007 \pm 0.0001$ | $0.0072 \pm 0.0002$ | $0.0070 \pm 0.0002$ | $98.6490 \pm 0.0388$ |
| Deq. ODE (I.S.) | $0.0017 \pm 0.0001$ | $0.0022 \pm 0.0002$ | $0.0189 \pm 0.0004$ | $0.0180 \pm 0.0004$ | $96.6150 \pm 0.0648$ |
| Deq. RealNVP (ELBO) | $0.0003 \pm 0.0001$ | $0.0004 \pm 0.0001$ | $0.0384 \pm 0.0010$ | $0.0283 \pm 0.0005$ | $95.1880 \pm 0.0771$ |
| Deq. RealNVP (I.S.) | $0.0003 \pm 0.0001$ | $0.0003 \pm 0.0000$ | $0.0208 \pm 0.0004$ | $0.0180 \pm 0.0004$ | $96.6340 \pm 0.0920$ |
| Man. ODE | $0.0012 \pm 0.0003$ | $0.0008 \pm 0.0002$ | $0.0098 \pm 0.0009$ | $0.0094 \pm 0.0007$ | $98.1780 \pm 0.1302$ |
| Möbius | $0.0027 \pm 0.0004$ | $0.0014 \pm 0.0003$ | $0.0542 \pm 0.0047$ | — | $88.7290 \pm 0.9332$ |

*Table 4.* Metrics of the dequantization algorithm in application to the orthogonal Procrustes problem and dequantization of a multimodal density on SO(3). When using the polar decomposition, results are averaged over ten independent trials for the multimodal distribution on SO(3) and nine independent trials for the orthogonal Procrustes problem; for the QR decomposition, results are averaged over nine trials.

| Experiment | Mean MSE | Covariance MSE | KL$(q\|p)$ | KL$(p\|q)$ | Relative ESS |
|---|---|---|---|---|---|
| Procrustes (ELBO - Polar) | $0.0021 \pm 0.0008$ | $0.0012 \pm 0.0005$ | $0.0193 \pm 0.0069$ | $0.0173 \pm 0.0053$ | $96.9489 \pm 0.7649$ |
| Procrustes (I.S. - Polar) | $0.0038 \pm 0.0020$ | $0.0015 \pm 0.0008$ | $0.0301 \pm 0.0126$ | $0.0202 \pm 0.0075$ | $95.6944 \pm 1.4654$ |
| Procrustes (ELBO - QR) | $0.0011 \pm 0.0003$ | $0.0008 \pm 0.0003$ | $0.0124 \pm 0.0032$ | $0.0095 \pm 0.0015$ | $97.9678 \pm 0.3325$ |
| Procrustes (I.S. - QR) | $0.0015 \pm 0.0005$ | $0.0011 \pm 0.0004$ | $0.0174 \pm 0.0072$ | $0.0122 \pm 0.0029$ | $96.6267 \pm 0.6326$ |
| SO(3) (ELBO - Polar) | $0.0007 \pm 0.0002$ | $0.0029 \pm 0.0003$ | $0.0443 \pm 0.0011$ | $0.0415 \pm 0.0059$ | $96.2930 \pm 0.0649$ |
| SO(3) (I.S. - Polar) | $0.0004 \pm 0.0001$ | $0.0014 \pm 0.0001$ | $0.0207 \pm 0.0028$ | $0.0235 \pm 0.0029$ | $97.7280 \pm 0.1136$ |
| SO(3) (ELBO - QR) | $0.0017 \pm 0.0004$ | $0.0054 \pm 0.0006$ | $0.0563 \pm 0.0060$ | $0.0363 \pm 0.0041$ | $93.5633 \pm 2.1331$ |
| SO(3) (I.S. - QR) | $0.0012 \pm 0.0004$ | $0.0020 \pm 0.0004$ | $0.0260 \pm 0.0017$ | $0.0219 \pm 0.0021$ | $94.3256 \pm 2.8099$ |

multimodal density on $\mathbb{S}^2$, we find that dequantization with an ambient neural ODE model is most effective, with ELBO maximization giving the smallest KL-divergence metrics. All dequantization algorithms out-performed the Möbius transformation on the sphere but only dequantization with an ambient ODE and ELBO minimization outperformed the manifold neural ODE method.

### 4.2. Orthogonal Group

The previous two examples focused on manifolds composed of spheres and circles. We now examine density estimation on the orthogonal group, where we consider inference in a probabilistic variant of the orthogonal Procrustes problem; we seek to sample orthogonal transformations that transport one point cloud towards another in terms of squared distance. We consider parameterizing a distribution in the

ambient Euclidean space using RealNVP in these experiments. Results are presented in table 4. We observe that optimizing the ELBO objective function (eq. (4)) tended to produce better density estimates than the log-likelihood (eq. (5)). Nevertheless, we find that either dequantization algorithm is highly effective at matching the target density.

We may also leverage corollary 1 so as to apply our method to the "dequantization" of SO(n). As an example, we consider a multimodal density on SO(3). Results of applying our method to sampling from this distribution are also shown in table 4. In this example we find that minimizing the negative log-likelihood using importance sampling tended to produce the best approximation of the first- and second-moments of the distribution, in addition to smaller KL-divergence metrics.

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
