# OpenReview forum: "Manifold Density Estimation via Generalized Dequantization"
_ICML.cc/2021/Workshop/INNF — INNF+ 2021 poster_

### Official Review · Reviewer_GEXT · 2021-06-07

**Rating:** Accept
**Confidence:** 3

**Summary:**

This work proposes to view density estimation on manifold structured data as a “quantization” problem; given observations on a manifold, the authors use auxiliary variables to lift the original observations to an Euclidean space (the dequantization step) where then flexible approaches, such as normalising flows, can be performed. The density on the manifold can then be obtained by marginalizing over the auxiliary variables (the quantisation step). The authors experimentally show that this approach compares favorably against the current state of the art.

**Justification For Rating:**

Overall I find the idea intuitive and reasonable. It is on topic for this workshop so I recommend for acceptance. Having said that, I believe that there are also a couple of things that the authors should address and expand upon. First of all, section 2 is missing references about the Theorems, Corollaries and other claims. Secondly, I believe the authors should also expand on the impact of the choice of the distribution over “z” on the performance of the algorithm; the authors could explain what it is and whether it has any free parameters that can be optimised. Finally, the notation needs a bit work and is not entirely consistent (e.g., compare eq. 3 with eq.4,5).

---

### Official Review · Reviewer_gptp · 2021-06-12

**Rating:** Accept
**Confidence:** 3

**Summary:**

This paper proposes an approach for density estimation on manifolds via "dequantization", or marginalizing out auxiliary manifold in the ambient Euclidian space. The authors use normalizing flows to model density in the ambient space. They compare the proposed approach experimentally to prior works on density estimation on the sphere, tori and orthogonal group.

**Justification For Rating:**

The paper proposes an interesting and, to the best of my knowledge, novel approach for density estimation in manifolds. The approach is based on the fact that for the ambient Euclidian space the coordinate system could be changed to the composition of the target manifold and auxiliary manifold. A general density estimator e.g. normalizing flows can be used to model density in the ambient space, and then by marginalizing out auxiliary space, one gets an approximated distribution on a manifold. Experimentally the proposed method outperforms two prior works in most settings.


**[Optional] Respond To Feedback Request By The Authors:**

I would suggest to add references to all Figures and Tables and discuss them in the main text (Figures 1-2 and Table 1 are not discussed).

Although the main ideas are clear, it would be helpful to give more details, explanations and intuition regarding the proposed approach, so the clarity of the paper could be improved. E.g. in the problem setting in section 3 lines 134-140, it would help to be explicit about how one has to choose $\mathcal{Z}$, $G$, $\pi_{\mathcal{X}}$ given initial manifold $\mathcal{Y}$, maybe an algorithm box would help. Also, which target distributions were used in the experiments?

For a longer version of the paper, it would be helpful to add background section and derivations of the theoretical claims.

---

### Official Review · Reviewer_UcV4 · 2021-06-13

**Rating:** Borderline Reject
**Confidence:** 3

**Summary:**

This paper proposes to use Normalizing Flows for estimating data density on structured manifolds. The idea is to define an invertible transformation from $\mathbb{R}^m$ to  $\mathbb{R}^p \times \mathbb{R}^n$, where $\mathbb{R}^n$ corresponds to the structured manifold of interested and $\mathbb{R}^p$ corresponds to the auxiliary variables that need to be marginalized out. The paper proposes to learn the invertible transformations as normalizing flows (RealNVP or Neural ODE) by optimizing either the ELBO or the log-likelihood (estimated via importance sampling).

**Justification For Rating:**

The paper considers an interesting use case for normalizing flows and a potentially worthwhile contribution to the literature on density estimation on structured manifolds. However, I have several concerns about the way it is presented.

* The proposed method seems to resemble a VAE a lot (e.g. with a flow-based encoder / decoder), however the connection to VAEs is not discussed in the manuscript. I believe it's important to draw similarities and differences from VAEs in this case, in particular with methods likes SurVAE Flows (D. Nielsen *et al.*, 2020). More generally, connections to related literature should be made explicit (e.g.  Mobius flows appear only in the Results section and are not mentioned before).
* The role of the dequantization distribution, and if / how it is related to the type of manifold considered (e.g. Table 1) was not very clear to me.
* Overall experimental details seem to be omitted from the paper - e.g. what kind of architectures were used, what were the dataset sizes, how many importance samples were taken, how were the models optimized, etc?
* Evaluation metrics should be explained more, e.g. what does $p$ and $q$ refer to in the KL-divergence metrics? Make it clear that Mean MSE and Covariance MSE refer to the first and second moments. Please also explain ESS. Beyond that. it is difficult to understand whether the results are strong based on these metrics. Guiding the readers through the tables would be helpful (e.g. boldface any clear "winner" methods).


*Minor comments*
* It is no obvious what "ambient space" refers to. Can this be made precise?
* I found that the introduction and abstract made more sense once I have read the examples in Sections 2.1 and 4. It could help an inexperienced reader if these examples are introduced early on.
* It wasn't clear to me what "projecting out any auxiliary structures" refers to. If it refers to marginalizing some of the variables, I am not certain that "projecting out" accurately captures this.
* "The torus appears the product manifold" - missing "the"?
* Is the definition of $x_i$ in equation (2) also meaningful when $x_i \in O_j$ and $i \neq j$?
* In equation (3) variable $\chi$ has not been defined, but presumably refers to $\mathbb{R}^m$ as before

---

### Decision · Program_Chairs · 2021-06-14

Accept (poster)